# Degradation-Suppressed Cocoonase for Investigating the Propeptide-Mediated Activation Mechanism

**DOI:** 10.3390/molecules27228063

**Published:** 2022-11-20

**Authors:** Nana Sakata, Ayumi Ogata, Mai Takegawa, Yuri Murakami, Misaki Nishimura, Mitsuhiro Miyazawa, Teruki Hagiwara, Shigeru Shimamoto, Yuji Hidaka

**Affiliations:** 1Faculty of Science and Engineering, Kindai University, 3-4-1 Kowakae, Higashi-Osaka 577-8502, Japan; 2Institute of Agrobiological Sciences, National Agriculture and Food Research Organization, Tsukuba 305-8634, Japan

**Keywords:** enzyme, folding, propeptide, trypsin, zymogen

## Abstract

Cocoonase is folded in the form of a zymogen precursor protein (prococoonase) with the assistance of the propeptide region. To investigate the role of the propeptide sequence on the disulfide-coupled folding of cocoonase and prococoonase, the amino acid residues at the degradation sites during the refolding and auto-processing reactions were determined by mass spectrometric analyses and were mutated to suppress the numerous degradation reactions that occur during the reactions. In addition, the Lys^8^ residue at the propeptide region was also mutated to estimate whether the entire sequence is absolutely required for the activation of cocoonase. Finally, a degradation-suppressed [K8D,K63G,K131G,K133A]-proCCN protein was prepared and was found to refold readily without significant degradation. The results of an enzyme assay using casein or Bz-Arg-OEt suggested that the mutations had no significant effect on either the enzyme activity or the protein conformation. Thus, we, herein, provide the non-degradative cocoonase protein to investigate the propeptide-mediated protein folding of the molecule. We also examined the catalytic residues using the degradation-suppressed cocoonase. The point mutations at the putative catalytic residues in cocoonase resulted in the loss of catalytic activity without any secondary structural changes, indicating that the mutated residues play a role in the catalytic activity of this enzyme.

## 1. Introduction

Zymogen proteins are produced and folded in the form of inactive precursor proteins. It is well known that the propeptide sequences in such precursor proteins play several roles, including serving as an inhibitor and as an intramolecular chaperone [1,2,3,4,5,6]. However, it is difficult to study the folding mechanism of a protease, especially propeptide-mediated folding, since numerous degradation reactions also occur during the refolding and the conversion of the enzyme to the active mature form. Therefore, to investigate the role of the propeptide in the precursor protein, these degradation reactions need to be eliminated.

Cocoonase (CCN), a protein produced by the silkworm (*Bombyx mori*), is known to specifically digest a sericin protein, a protein that coats the fibroin of a silkworm cocoon, and belongs to a trypsin-like protease family [5,7,8,9]. Cocoonase is synthesized as an inactive zymogen precursor protein (proCCN) and is auto-catalytically or enzymatically processed into the bioactive mature form, cocoonase, on the release of the N-terminal 12-residue peptide, as shown in Figure 1 [10]. We recently proposed that the CCN protein is kinetically trapped at the molten globule state in the folding pathway without the propeptide region and shifts to the native conformation with the assistance of the propeptide region as an intramolecular chaperone at the final step of the folding pathway [11].

To validate this hypothesis, refolding reactions were performed under several conditions. However, it was difficult to obtain folding intermediates since numerous degradation reactions were observed during the refolding and the enzyme activation, as well as that of trypsin. Therefore, to investigate the folding and stabilization of the CCN and proCCN protein, we prepared some degradation-suppressed proCCN or CCN mutant proteins.

For this purpose, the degradation sites during the refolding and auto-processing reactions were determined by mass spectrometric analyses and proteins in which the putative degradation sites were mutated were prepared (Appendix A). A triple mutation at the putative degradation sites significantly decreased the extent of the degradation that occurred during the activation by the auto-processing, resulting in an improved recovery of the active mature enzyme. In addition, to investigate the role of the entire propeptide sequence in the folding of cocoonase, the Lys^8^ residue at the propeptide sequence was also mutated and the folding and enzyme activity of this product were examined.

Cocoonase shares a high homology with trypsin (Appendix A) [10]. Based on a homology alignment, the putative catalytic residues are assigned to the His^56^, Asp^99^, and Ser^193^ residues. However, this has not been confirmed by experimental data. In addition, it is also noteworthy that the folding mechanism of cocoonase has not been studied when the catalytic residues are mutated to inactive residues to eliminate the nonspecific degradation that occurs during the refolding reaction. Therefore, to determine the catalytic residues, we prepared mutant proteins in which the putative catalytic residues were mutated in addition to the degradation-suppressed mutations and estimated the enzyme activity and conformation of these molecules.

## 2. Results and Discussion

To suppress the degradation that occurs during the refolding and activation reactions, the sites on the molecules responsible for the degradation were explored. First, the wild type CCN protein, which was purified with an octadecylsilyl (ODS) open column (Nacalai tesque, Kyoto, Japan), was refolded and auto-processed by the previously described method [11]. Several degradation products, including an 18 kDa protein, were produced, as shown in Figure 2a. Therefore, to explore the cleavage sites during the reactions, the components of the reaction mixtures were separated by Reversed-Phase High Performance Liquid Chromatography (RP-HPLC), as shown in Figure 3a. The major peaks 1 and 2 in Figure 3a were purified and subjected to mass spectrometric analyses, providing the mass values of 25,243 Da and 17,872 Da, respectively. This result indicated that peaks 1 and 2 correspond to the proCCN protein and the C-terminal fragment (Val^64^ to Thr^235^), respectively, indicating that the proCCN protein is cleaved at the C-terminal peptide bond of the Lys^63^ residue. The mass value of the peak 2 protein in Figure 3a was also compatible with the apparent molecular weight of the degradation product (18 kDa protein) recorded using the SDS-PAGE analysis shown in Figure 2a. Therefore, to suppress this degradation at position 63, we prepared the recombinant [K63G]-proCCN protein, in which the Lys^63^ residue was replaced by a Gly residue. As expected, the 18 kDa protein was not produced during the refolding and auto-processing reactions of the [K63G]-proCCN protein, as shown in Figure 2b. However, a new degradation product, a 6 kDa protein in Figure 2b, was produced during the activation of the [K63G]-proCCN protein. Therefore, to further explore the cleavage sites of the [K63G]-proCCN protein, the reaction mixtures of the [K63G]-proCCN protein were also applied to RP-HPLC, and the degradation products were separated, as shown in Figure 3b. Peak 3 in Figure 3b provided mass values of 5789 Da and 5559 Da on the mass spectrometric analyses, indicating that the degradation fragments (Thr^132^ to Lys^186^ and Leu^134^ to Lys^186^) were produced by the cleavage at the -Lys^131^-Thr^132^- and the -Lys^133^-Leu^134^- sequences during the refolding and auto-processing reactions. As described in the introduction, cocoonase shares a high homology with trypsin [10]. It is known that the peptide bond between the Lys^138^ and Cys^139^ residue of human trypsin is cleaved during the refolding and auto-activation reactions [12]. This position is relatively the same as the -Lys^133^-Leu^235^- sequence of cocoonase. Therefore, to avoid nonspecific cleavages at those sites, we introduced another double mutation to the [K63G]-proCCN molecule. For this purpose, [K63G,K131G,K133A]-proCCN, referred to as proCCN’, was prepared, refolded, and auto-processed. The level of degradation products was significantly decreased during the refolding and auto-processing reactions, as shown in Figure 2c. The overall yields of the refolding and auto-processing reactions of the wild type CCN, [K63G]-proCCN, and proCCN’ proteins were approximately 9%, 13%, and 59%, respectively. It is also known that the refolding recoveries of trypsin and trypsinogen are also extremely low since significant amounts of aggregation and degradation also occur during this process. Indeed, the refolding recovery of trypsin from the reduced form was reported to be 12% [13,14]. To solve this problem, the refolding of trypsin was carried out in an immobilized state or on an inhibitor-immobilized resin [14]. In addition, the folding recovery of trypsinogen was only 50%, even in the case of the immobilized state [15]. As shown in our study, the triple mutation at the Lys^63^, Lys^131^, and Lys^133^ residues of cocoonase significantly increased the recovery of the refolding and auto-processing reaction without the need for inhibitors.

Thus, we were able to obtain relatively high levels of degradation-suppressed cocoonase. However, a certain amount of degraded protein was still observed between the proCCN and CCN protein band for the auto-processing reactions, as shown in lanes 6 and 7 in Figure 2c and lane 1 in Figure 2d, implying that the propeptide region was cleaved at the Lys^8^ residue. To investigate the role of the propeptide region, the entire propeptide sequence should be maintained during the refolding reaction. Therefore, to address this issue and eliminate further degradation, the Lys^8^ residue should be mutated. For this purpose, the Lys^8^ residue was replaced with an Asp residue, considering the effects for the intramolecular function of the propeptide sequence. As described in the introduction, trypsinogen shares a high homology with prococoonase and possesses the enterokinase recognition sequence (DDDDK), in which the Asp residue is located at relatively the same position as the Lys^8^ residue of prococoonase, in place of the propeptide sequence (KDEEK) of cocoonase, as shown in Appendix A. Therefore, the [K8D]-proCCN’ protein was recombinantly expressed, refolded, and auto-processed. The band corresponding to the degradation protein between the proCCN and CCN mutants was no longer detected in the SDS-PAGE analyses, as shown in lanes 3 and 4 in Figure 2d. Thus, we finally obtained the degradation-suppressed mutant protein, [K8D]-proCCN’, for use in our investigations of the folding mechanism of prococoonase and cocoonase.

To confirm the protease activity of the mutant proteins, we assayed the protease activities of the recombinant proteins using *N*^α^-Benzoyl-_L_-arginine ethyl ester (Bz-Arg-OEt) as a substrate [16,17]. As summarized in Table 1, the protease activities of the mutant proteins were compatible with that of the wild type CCN protein, indicating that the triple mutation had no effect on the enzyme activities of these proteins.

To further characterize the mutant proteins prepared above, secondary structural analyses were performed by collecting circular dichroism (CD) spectra. As shown in Figure 4, the CD spectra of the wild type and the mutant proteins were completely overlapped, indicating that the above mutations had no effect on the secondary/tertiary structure of the cocoonase proteins. Importantly, replacing the Lys^8^ residue with an Asp residue at the propeptide region did not affect the conformations of the mutant proteins or the enzyme activity after the processing, indicating that the Lys residue is not an essential residue for kinetic assistance as an intramolecular chaperone function of the propeptide region to accelerate the reaction from the molten globule to the native structure. In the case of a homologous protein, trypsin, the propeptide region is not required for the final enzyme activity, and it simply inhibits the enzyme activity of the protein after the folding has occurred [13,14]. Contrary to the previous report, it was also reported that the artificial propeptide sequences dramatically affected the final conformation and enzyme activity of trypsin [18], suggesting that the propeptide region essentially interacts with the active site during the refolding reaction. However, it is difficult to determine the role of the propeptide region in the refolding reaction of trypsin because a variety of other degradations occur during the refolding reaction because of the strong enzyme activity of trypsin [13]. In our study, we were able to successfully prepare the degradation-suppressed mutant protein. Utilizing this mutant protein, it will be possible to examine the role of the propeptide region in the folding mechanism of not only cocoonase, but also of proteins in the trypsin super family. Thus, this research provides a practical tool for investigating the folding mechanism of the trypsin super family.

As describe above, cocoonase is a member of a serine protease family of proteins and possesses a high homology with trypsin. However, the catalytic residues responsible for the protease activity of this protein have not been determined as of this writing. It would be of interest to investigate the propeptide-mediated folding of cocoonase if the catalytic residue was precisely known and mutated to an inactive residue, which would result in the suppression of the degradation of the folding intermediates. Therefore, to examine the residues in the catalytic triad of cocoonase that determine its activity, the putative catalytic residues (His^56^, Asp^99^, and Ser^193^ residues) were also mutated to an Ala residue, based on a homology alignment between cocoonase and trypsin [19], as shown in Appendix A. For this purpose, the triple mutant protein (CCN’) was employed as a template protein for the mutation to avoid nonspecific degradation that might occur during the refolding and processing reactions. Thus, the recombinant [H56A]-proCCN’, [D99A]-proCCN’, and [S193A]-proCCN’ proteins were designed, expressed in *E. coli* cells, and were found to refold well (data not shown). After the purification of the recombinant [H56A]-proCCN’, [D99A]-proCCN’, and [S193A]-proCCN’ proteins, CD measurements were carried out to estimate whether the mutation affected the conformations of these proteins. The CD spectra of all of the mutant proCCN’ proteins were essentially overlapped with that of the proCCN’, as shown in Figure 5a, indicating that the proteins had folded properly, as well as the wild type proCCN and proCCN’. As assumed, no auto-processing reactions were observed for these mutant proteins, as shown in Figure 6. Therefore, the propeptide regions were cleaved by a trypsin treatment. However, it was not possible to measure the enzyme activities of the mature forms of the mutant protein after the trypsin treatment since trypsin could not be completely removed from the reaction mixtures even by using immobilized trypsin agarose (ProteoChem, Inc., Hurricane, UT, USA) [20], or after purification by cation-exchange chromatography (data not shown). Therefore, to obtain the mature forms of the mutant proteins, in which the active sites had been replaced by the Ala residues, we introduced an enterokinase recognition sequence (DDDDK) at the processing site of the proCCN’ mutant proteins. The cassette mutation from the -Lys-Asp-Glu-Glu- to the -Asp-Asp-Asp-Asp- sequence in cocoonase was originally introduced in an *E. coli* expression system [21]. The mature cocoonase protein obtained using the enterokinase recognition sequence showed a similar *k*_cat_ value to that of trypsin, although the *K*_m_ value was not determined [21]. Therefore, the cassette mutation does not essentially affect the catalytic activity of cocoonase. Thus, [K8D,E10D,E11D,H56A]-proCCN’, [K8D,E10D,E11D,D99A]-proCCN’, and [K8D,E10D,E11D,S193A]-proCCN’ were designed, expressed in *E. coli* cells, and refolded, and the conformations of the molecules were confirmed by CD measurements, as shown in Figure 5b. The CD spectra of all of the mutant proteins were nearly the same as that of proCCN’, indicating that the protein conformations of the mutants were accurately constructed, as well as that of proCCN. Thus, the mutant proCCN’ proteins with the cassette mutation at the propeptide region were prepared and further processed to the mature forms.

The mutant proCCN’ proteins were treated with enterokinase, yielding the mature mutant proteins, [H56A]-proCCN’, [D99A]-proCCN’, and [S193A]-proCCN’, as shown in Figure 7a–c. The mature CCN’ mutant proteins were then treated with an enterokinase capture kit according to the manufacture’s recommendations. The resulting mutant CCN’ proteins did not show any protease activity on the casein zymography assay while the CCN’ proteins that were auto-processed from proCCN’ or [K8D]-proCCN’ showed active protease bands (white bands) on the gels, as shown in Figure 8a. The active protease bands of the CCN’ proteins in lane 1 in Figure 8a (upper in the left column) were also observed before the auto-processing reaction since trace amounts of the mature forms had already been produced before the reaction. In addition, an enzyme assay using Bz-Arg-OEt clearly indicated that the mutant CCN’ proteins, in which the catalytic residues had been mutated, did not possess protease activities, as shown in Figure 8b. The *in silico* analyses of the cocoonase using Alphafold2 suggested that the putative residues form a catalytic triad at relatively the same positions as that of trypsin (PDB ID: 5T3H), as shown in Appendix A [22]. Therefore, we conclude that the His^56^, Asp^99^, and Ser^193^ residues are the actual participants in the catalytic triad.

In this study, several amino acid residues were mutated to suppress the degradation that occurs during the refolding and auto-processing reaction of cocoonase. We previously reported that the propeptide region assists in the final stage of the folding pathway from the molten globule to the native form [11]. Similar phenomena were also reported for the folding of chymotrypsinogen [1]. In addition, the role of the propeptide-region of the trypsin family in the folding process still remains obscure because numerous degraded proteins are produced in the refolding reaction. The *in silico* structural analyses of trypsin detected interactions between the propeptide region and the catalytic region, as we predicted [18]. Therefore, the degradation-suppressed mutant cocoonase prepared in this paper should be utilized to investigate the disulfide-coupled folding pathway of cocoonase as well as other members of the serine protease family of proteins.

Moreover, cocoonase is also expected to be utilized industrially in the refining process of native silk because the boiling method that is currently used for refining silk damages the silk protein and diminishes the degree of the glistening of the final product [10]. The degradation-suppressed cocoonase described in this study will be useful for treating cocoons under the much milder conditions and provide non-damaged silk fibrils. However, we estimate that the conditions used for the storage of cocoonase for industrial usage continues to be a problem since the recombinant cocoonase is prone to undergoing aggregation after the cleavage of the propeptide region, as we reported previously [11]. Cocoonase possesses only three intramolecular disulfide bonds, indicating that it lacks two disulfide bonds, compared to that of human trypsin (5 disulfide bonds). This lower number of the disulfide bonds may cause the cocoonase molecule to be structurally unstable. Therefore, additional disulfide bonds may be required for producing a conformationally stable form of cocoonase for storage and industrial usage.

In conclusion, mutations at the Lys^8^, Lys^63^, Lys^131^, and Lys^133^ residues of cocoonase significantly suppressed the degradation of the protein that occurs during the refolding and auto-processing reaction of this protein. In addition, the His^56^, Asp^99^, and Ser^193^ residues were determined to be the catalytic residues of cocoonase. Utilizing mutant proteins for investigating the disulfide coupled-folding of cocoonase will provide new insights into the folding process of not only cocoonase, but also other members of the trypsin super family.

## 3. Materials and Methods

### 3.1. Materials

Glutathione and Bz-Arg-OEt were purchased from the Peptide Institute, Inc. (Osaka, Japan). Casein was purchased from Fuji Film Wako Pure Chemicals, Ltd. (Osaka, Japan). All chemicals and solvents used were of reagent grade. The cDNA encoding proCCN (Appendix A) was prepared by Eurofins Japan (Tokyo, Japan), and its codon bias was optimized for an *E. coli* expression system using the manufacture’s program GENEius (http://www.geneius.de/; accessed on 13 May 2017). Protein solutions and *E. coli* cells were centrifuged using CF15RXII (HITACHI, Tokyo, Japan).

### 3.2. Construction of the Expression Vectors of the Recombinant Mutant Proteins in E. coli Cells

The primer sequences used in this study were summarized in Appendix A. The cDNA of the proCCN mutant proteins, [K63G]-proCCN, [K63G,K131G,K133A]-proCCN, and [K8D,K63G,K131G,K133A]-proCCN, were prepared by PCR, using the synthetic cDNA encoding proCCN as a template [11]. The PCR reaction was carried out using Platinum Pfx DNA polymerase (Invitrogen, Thermo Fisher Scientific, Inc., Waltham, CA, USA). The resulting amplified cDNA’s were then subcloned into the pET-17b expression vector (Novagen, Glendale, CA, USA), following introduction of an *Nde*I and a *Xho*I site at its 5′ and 3′ end, respectively. The resulting expression vectors, referred to as pMT1, pMT2, and pNS1, contained the cDNA of [K63G]-proCCN, [K63G,K131G,K133A]-proCCN, and [K8D,K63G,K131G,K133A]-proCCN, respectively. The cDNA sequences of the vectors were confirmed by the Eurofins Japan DNA sequencing service (Tokyo, Japan).

The cDNA of the mutant proteins, in which the putative catalytic residue is mutated to the Ala or Gly residue, were also prepared by PCR, using the pMT2 vector as a template. The vector constructions were carried out in the same manner as described above. The resulting expression vectors, referred to as pAO4, pAO5, pAO6, and pNS1, contained the cDNA of [H56A,K63G,K131G,K133A]-proCCN, [K63G,D99A,K131G,K133A]-proCCN, and [K63G,K131G,K133A,S193A]-proCCN.

The cDNA sequences of the cassette mutant proteins, in which the propeptide sequence (KDEEK) is cassette-mutated to the DDDDK sequence, were also prepared by PCR, using the pAO4, pAO5, and pAO6 vectors as templates. The construction of the vectors was carried out in the same manner as described above. The resulting expression vectors, referred to as pAO7, pAO9, and pAO10, contained the cDNA of [K8D,E10D,E11D,H56A,K63G,K131G,K133A]-proCCN, [K8D,E10D,E11D,K63G,D99A,K131G,K133A]-proCCN, and [K8D,E10D, E11D,K63G,K131G,K133A,S193A]-proCCN, respectively. The cDNA sequences of all the vectors constructed in this study were confirmed by analyses using the dideoxy method.

### 3.3. Protein Expression and Purification of Recombinant Proteins

The protein expression was performed following a previously reported method [23,24]. All of the mutant proteins were obtained as inclusion bodies in *E. coli* cells and possessed a Met residue at the N-terminus of the proCCN mutant proteins derived from the *Nde*I site during the subcloning. After sonication of the cells, the mixtures were centrifuged at 15,000 rpm for 15 min. The residues were subsequently washed with 0.5% Triton X-100, water, 80% CH_3_CN/0.05% TFA. The final products were resuspended in Milli Q water and the protein levels were estimated by SDS-PAGE analyses using BSA as a standard protein.

### 3.4. Refolding Reactions of Recombinant Proteins and the Enzymatic Activation of proCCN

The refolding reactions of the reduced forms of the recombinant mutant proCCN proteins were carried out by a previously described method with minor modifications [11]. Briefly, the recombinant proteins were solubilized through treatment with 6 M Gu/HCl in 20 mM Tris/HCl (pH 8.0) containing 10 mM dithiothreitol (DTT) at 50 °C for 30 min. After centrifugation, to completely remove the DTT, the protein solution was dialyzed twice against 4 M urea/0.05% TFA containing 0.5 M NaCl, and the resulting solution was centrifuged at 15,000 rpm for 15 min. The supernatants were mixed with an equal volume of 0.2 M Tris/HCl containing 0.5 M NaCl, 4 mM DPAET, and 0.4 mM GSSG and allowed to stand for 2 h at 4 °C. The reaction mixtures were subsequently diluted with an equal volume of 0.2 M Tris/HCl containing 0.5 M NaCl (the final concentration of urea, 1 M) and allowed to stand for 2 days at 4 °C.

The refolded proCCN was dialyzed twice against 50 mM sodium phosphate buffer (100 mL, pH 7.0) containing 20 mM NaCl. The resulting solution was then subjected to cation exchange chromatography, as described below.

### 3.5. Cation Exchange Chromatography

The proCCN and the CCN mutant proteins were purified by cation exchange chromatography using the ÄKTA purifier (GE Healthcare Japan, Tokyo, Japan). Briefly, the refolded proteins were dialyzed against 50 mM phosphate buffer (pH 7.0) containing 20 mM NaCl. After dialysis, the enzyme solution was applied to a HiTrap SP HP column (5 mL, GE healthcare Japan, Tokyo, Japan) that had been pre-equilibrated with buffer A (50 mM phosphate buffer, pH 7.0) and eluted with buffer B (50 mM phosphate buffer, containing 1 M NaCl, pH 7.0). The concentrations of the eluted proteins were determined by the Bradford’s method or by UV absorbance.

### 3.6. Activation of the Recombinant Prococoonase by Treatment with Enterokinase

The fractions of the proCCN mutant proteins obtained by cation exchange chromatography were approximately concentrated to solutions containing 1 mg/mL of protein using Amicon Ultra-15 (10K, Merck Millipore, Darmstadt, Germany) at 4 °C. For auto-processing for enzyme activation, the temperature was shifted to 25 °C, and the reactions were stopped by freezing at each reaction time [11]. The reaction solutions were then analyzed by SDS-PAGE.

In the case of processing by enterokinase, the proCCN mutant proteins (0.1 mg of protein) were treated overnight with an enterokinase solution (1 mg/mL, 15 μL) at 25 °C. The reaction mixtures were further treated with EKapture^TM^ agarose (50 μL) pre-equilibrated with 0.2 M Tris/HCl (pH 7.4), 20 mM CaCl_2_, 0.5 M NaCl, and centrifuged at 10,000 rpm for 1 min at 4 °C. The supernatants were analyzed by SDS-PAGE.

In addition, the prepared proteins were characterized by mass spectrometric and amino acid analyses (data not shown). Aliquots of the protein solutions were lyophilized and stored at 4 °C until used.

### 3.7. Casein Zymography

The protease activities of the recombinant proteins were estimated by casein zymography [25]. Briefly, the refolded proteins were electrophoresed on an SDS-substrate 15% polyacrylamide gels containing casein (1 mg/mL). Samples were mixed with the same volume of SDS sample buffer (50 mM Tris/HCl, 4.5% SDS, 9% glycerol, 0.2% bromophenol blue, pH 7.0), though not boiled, and were then loaded and separated at room temperature. After electrophoresis, the SDS was removed by shaking the gels twice in 2.5% Triton X-100 for 30 min, followed by another 90 min of shaking in 50 mM Tris/HCl (pH 8.0) for the enzyme reaction at room temperature. The gels were stained with coomassie blue R250 (60 mg/L) in 10% isopropanol with 10% acetic acid and destained in water.

### 3.8. Determination of Enzyme Activity Using Bz-Arg-OEt

The esterase activity of the recombinant enzymes was estimated according to a previously reported method [16,17]. Briefly, the enzymatic reactions were carried out in 50 mM Tris/HCl (1.0 mL, pH 8.0) containing 25 mM NaCl and 1 mM EDTA as follows: the purified enzyme (approximately 1 μmol/10 μL) was mixed with 990 μL of Bz-Arg-OEt (final concentration: 25–100 μM) and then incubated at 25 °C for 5 min. The absorbance at 253 nm was measured every 5 sec for a period of 5 min. The initial velocity was plotted as a function of the substrate concentration, and the apparent kinetic parameters, *K*_m_ and *V*_max_, were calculated by fitting the experimental points to a Lineweaver–Burk plot [26].

### 3.9. Reversed-Phase High Performance Liquid Chromatography (RP-HPLC)

The High Performance Liquid Chromatography (HPLC) apparatus was comprised of a HITACHI ELITE LaChrom system (L2130, HITACHI, Tokyo, Japan) or a Waters M600 multisolvent delivery system (Waters, Milford, MA, USA) equipped with a Hitachi L-3000 detector and a D-2500 chromatointegrator. Peptides and proteins were separated by RP-HPLC using a Cosmosil 5C_18_-AR-II column (4.6 × 150 mm, Nacalai tesque, Inc., Kyoto, Japan) and confirmed by MALDI-TOF/MS analyses [27].

### 3.10. Matrix-Assisted Laser Desorption/Ionization Time of Flight Mass Spectrometry (MALDI-TOF/MS)

The molecular masses of peptides and proteins were determined by means of an AXIMA confidence spectrometer (SHIMADZU Co., Kyoto, Japan) in the positive ion mode [27]. Mass spectrometric analyses of proteins and peptides were carried out in the linear modes using 3,5-dimethoxy-4-hydroxycinnamic acid (Tokyo Chemical Industry Co., Ltd., Tokyo, Japan) as matrices, respectively. In a typical run, the lyophilized sample (ca. 0.1 nmol) was dissolved in 0.05% TFA aq/50% CH_3_CN (1 μL), mixed with 1 μL of a matrix solution (10 mg/mL), and air-dried on the sample plate for use in MALDI-TOF/MS.

### 3.11. CD Measurements

CD spectra were recorded on a JASCO J720 spectrometer (JASCO Corporation, Tokyo, Japan) at 25 °C. The proteins that had been purified by cation exchange chromatography were dialyzed against 20 mM sodium phosphate buffer (pH 7.4) containing 20 mM NaCl, and their concentrations were determined by UV absorbance [11].

## Figures and Tables

**Figure 1 molecules-27-08063-f001:**
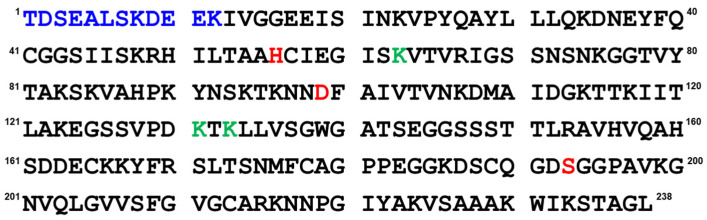
Amino acid sequence of prococoonase (proCCN). The auto-processing occurs between the K^12^ and I^13^ residues. The propeptide sequence, the mutation sites for the degradation, and the putative catalytic residues are indicated as blue, green, and red letters, respectively.

**Figure 2 molecules-27-08063-f002:**
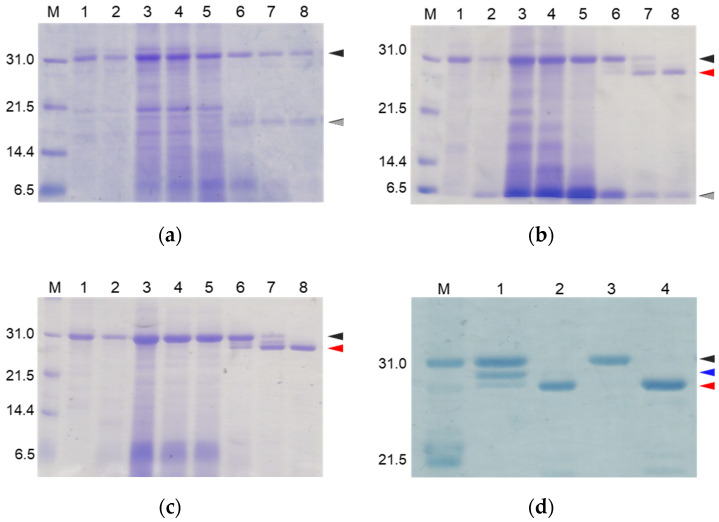
SDS-PAGE analyses of reaction solutions of the auto-processing of proCCN (**a**), [K63G]-proCCN (**b**), proCCN’ (**c**), and [K8D]-proCCN’ (**d**). Lanes 1 and 2: 0 min and 16 h at 4 °C, respectively; lanes 3–8: 0 min, 1 h, 3 h, 24 h, 48 h, and 72 h at 25 °C, respectively (**a**–**c**). Lanes 1 and 2: 0 min and 24 h of proCCN’, respectively; lanes 3 and 4: 0 min and 24 h of [K8D]-proCCN’, respectively (**d**). Black, gray, and red arrowheads indicate proCCN proteins, degradation products, and CCN proteins, respectively. Blue arrowhead indicates the degradation product that is cleaved at the Lys^8^ residue.

**Figure 3 molecules-27-08063-f003:**
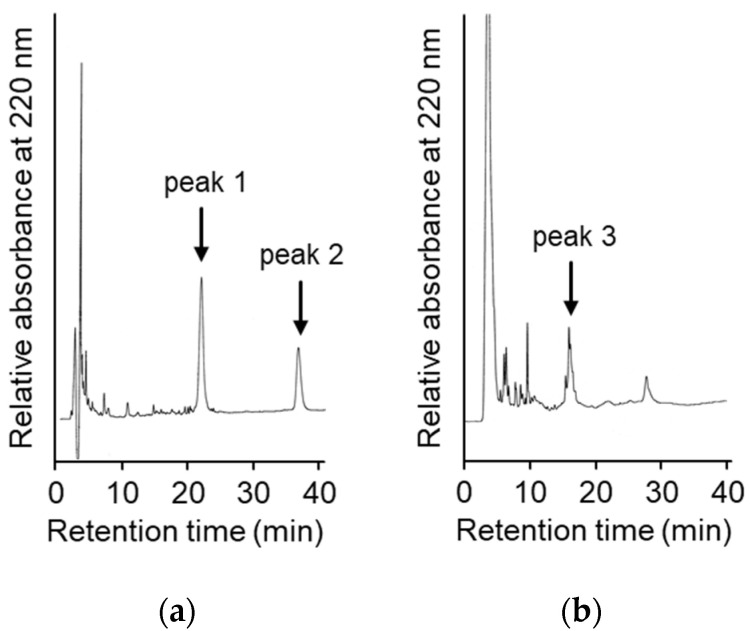
High Performance Liquid Chromatography (HPLC) profiles of the reaction solutions of the auto-processing of the recombinant proCCN (**a**) and [K63G]-proCCN (**b**). The reaction solutions at lanes 8 in Figure 2a,b were purified by RP-HPLC (**a**,**b**), respectively. Waters and Hitachi HPLC systems were used to analyze the reaction solutions.

**Figure 4 molecules-27-08063-f004:**
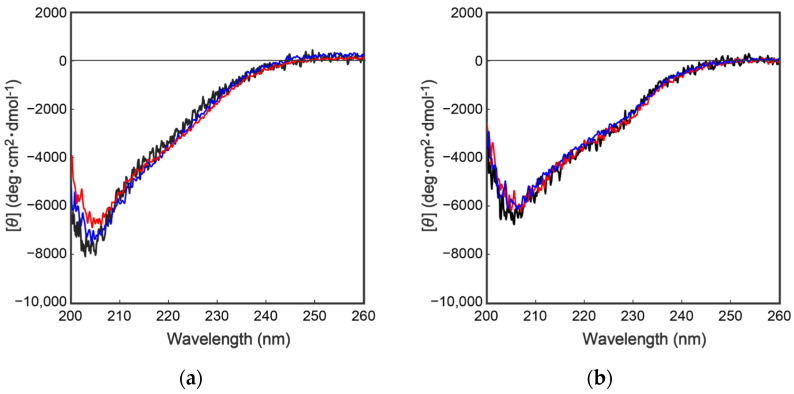
CD spectra of a series of proCCN (**a**) and CCN proteins (**b**). The black, blue, and red represent the CD spectrum of the wild type proCCN, [K63G,K131G,K133A]-proCCN and [K8D,K63G,K131G,K133A]-proCCN (**a**), and their mature form (**b**), respectively.

**Figure 5 molecules-27-08063-f005:**
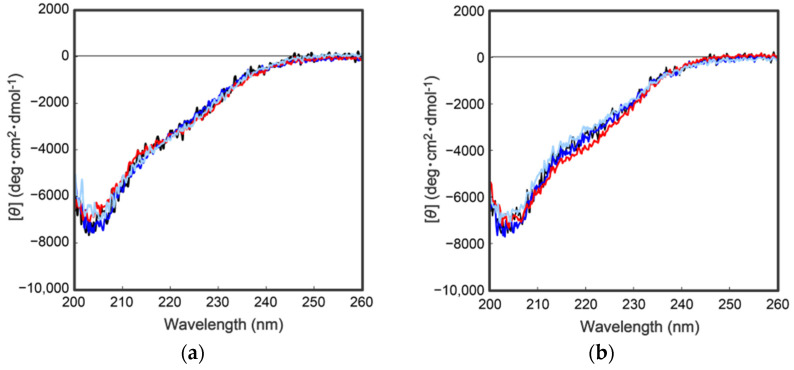
CD spectra of a series of proCCN (**a**) and the cassette mutated proCCN proteins (**b**). The black, blue, red, and light blue represent the CD spectrum of proCCN’, [H56A]-proCCN’, [D99A]-proCCN’, and [S193A]-proCCN’, respectively.

**Figure 6 molecules-27-08063-f006:**
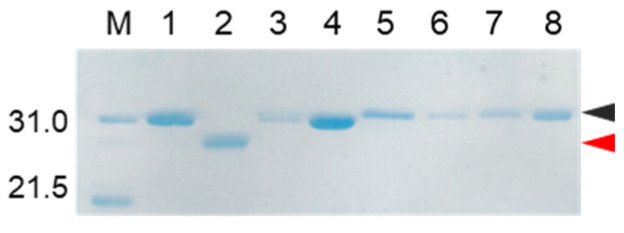
SDS-PAGE analyses of reaction solutions of the auto-processing of [H56A]-proCCN’, [D99A]-proCCN’, and [S193A]-proCCN’ proteins. Odd and even lanes: solutions before and after the reactions, respectively. Lanes 1 and 2, lanes 3 and 4, lanes 5 and 6, and lanes 7 and 8 represent the profiles of proCCN’, [H56A]-proCCN’, [D99A]-proCCN’, and [S193A]-proCCN’ proteins, respectively. Black and red arrowheads indicate the precursor and the mature form, respectively.

**Figure 7 molecules-27-08063-f007:**
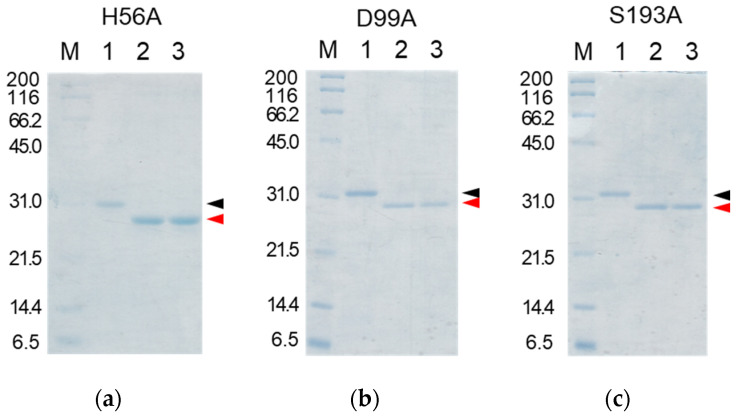
SDS-PAGE analyses of the reaction solutions of the auto-processing of the cassette mutated [H56A]-proCCN’ (**a**), [D99A]-proCCN’ (**b**), and [S193A]-proCCN’ (**c**) proteins. Lanes 1, 2, and 3: purified proCCN’, treated with enterokinase, and subsequently treated with the EKapture^TM^ agarose, respectively. Black and red arrowheads indicate proCCN and CCN protein, respectively.

**Figure 8 molecules-27-08063-f008:**
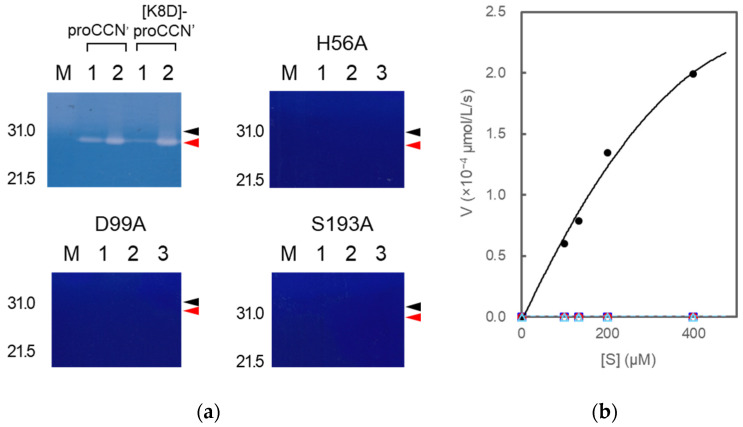
Casein zymography (**a**) of proCCN’, [K8D]-proCCN’, the cassette mutated [H56A]-proCCN’, [D99A]-proCCN’, and [S193A]-proCCN’ protein and plots of initial velocity vs. substrate concentration (**b**) for trypsin and the mutant proteins. Lanes 1 and 2 in the cases of proCCN’ and [K8D]-proCCN’: solutions before and after auto-processing, respectively (corresponding to Figure 2d); lanes 1, 2, and 3 in the cases of the cassette mutated [H56A]-proCCN’, [D99A]-proCCN’, and [S193A]-proCCN’ protein: purified proCCN’, treated with enterokinase, and subsequently treated with EKapture^TM^ agarose, respectively. Black and red arrowheads indicate proCCN and CCN protein, respectively. Trypsin was used as a standard enzyme for the control experiments in the enzyme assay using Bz-Arg-OEt. The black (closed circle), blue (open square), red (open circle), and light blue (open triangle) colors represent the data plots of trypsin, [H56A]-proCCN’, [D99A]-proCCN’, and [S193A]-proCCN’, respectively.

**Table 1 molecules-27-08063-t001:** The *K*_m_ and *V*_max_ values of the mutant proteins.

Recombinant Protein	*K*_m_ (M)	*V*_max_ (μmol/L/s)
Wild type CCN	6.4 × 10^−4^	4.5 × 10^−4^
[K63G]-CCN	4.0 × 10^−4^	1.0 × 10^−4^
[K63G,K131G,K133A]-CCN (CCN’)	3.7 × 10^−4^	1.4 × 10^−4^
The mature CCN’ derived from [K8D,K63G,K131G,K133A]-proCCN	9.2 × 10^−4^	4.6 × 10^−4^

The enzyme activity of the wild type is cited from a previous report [8]. The *K*_m_ and *V*_max_ values were calculated by fitting the experimental data points to a Lineweaver–Burk plot.

## Data Availability

Not applicable.

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
