# Peer review of "Degradation-Suppressed Cocoonase for Investigating the Propeptide-Mediated Activation Mechanism"

_molecules, 2022, doi:10.3390/molecules27228063_

Round 1

Reviewer 1 Report

In this study, to investigate the role of the propeptide sequence on the disulfide-coupled folding of cocoonase and prococoonase, the amino acid residues at the degradation sites during the refolding and auto-processing reactions were determined by mass spectrometric analyses and were mutated in order to suppress the numerous degradation reactions that occur during the reactions. It is an interesting study, it may be accepted for publication after minor revisions.

1, The figure 1 can be provided in supplementary material. The quality (resolution) of figures can be improved.

2, A schematic diagram to summarize the content of present study can be added in the introduction section.

3, There is no any signal can be seen in Figure 8.

4, Please indicate in the manuscript that how to calculate the Km and Vmax.

5, Please check and revise some minor problems. For example, there should be a space between he numeral and unit.

Author Response

Our reply to the specific comments is as follows:

  1. The reviewer recommended that Fig. 1 be moved to the supplementary material. However, we believe that the paper is much more readable as it is now. Therefore, please accept our proposal to put the figure in the main body.
    In addition, the quality of figures was improved, especially for Fig. 3.
  1. As the reviewer recommended, the schematic drawing to summarize the contents of the present study was added as a supplementary figure (Fig. S1) in the introduction. In addition, all supplementary figures were renumbered.
  2. As pointed it out by the reviewers, the original Fig. 8 was not suitable to sufficiently explain our conclusion. Therefore, we prepared a new Fig. 8 and added the control experiments using the active enzyme, the CCN’ proteins that had been auto-processed from proCCN’ and [K8D]-proCCN’. In addition, an enzyme assay using Bz-Arg-OEt as a substrate was also added to make our results more clear. In the casein zymography assay, the active enzyme is observed as a white band while the inactive enzyme is not observed on the blue gel.
    We also added additional text to explain the results of the experiments in Fig. 8 in the revised manuscript.
  1. Added in the Table 1. The Km and Vmax values were calculated by fitting the experimental data points to a Lineweaver-Burk plot.
  2. All typos and grammatical errors were addressed in the revised paper.

Reviewer 2 Report

The manuscript “Degradation-suppressed cocoonase for investigating the pro-peptide-mediated activation mechanism” [molecules-2025642-peer-review-v1] written by Nana Sakata, Ayumi Ogata, Mai Takegawa, Yuri Murakami, Misaki Nishimura, Mitsuhiro Miyazawa, Teruki Hagiwara, Shigeru Shimamoto and Yuji Hidaka describes investigations of the degradation of cocoonase that occurs during the refolding and auto-processing reaction. The authors identify lysine residues (Lys8, Lys63, Lys131 and Lys133) to be involved in the degradation process. Furthermore, they identify His56, Asp99, and Ser193 residues as the main catalytic residues of the catalytic center. In particular mutations and concomitant folding and activity measurements have been performed.

All experiments and calculations are performed with modern and common state of the art methods. The overall work is quite well planned and performed. All examinations appears sensible and lead to consistent results. These reported results and conclusions sound perspicuous and are well described. They possess some importance in furthering our knowledge about cocoonase and prococoonase, their folding and the active center. The manuscript is hence of interest in the fields of Biochemistry, Protein Chemistry and Natural Product Chemistry. It is worth publishing and offers a very good basis for further developments in this field.

However, one minor improvement might be made in the description and presentation before the manuscript can be published in “Molecules”. Therefore, there is one minor comments, which should be taken into account by the authors prior to acceptance of the manuscript:

Minor comments:

1)
The representations in Figure 8 are unambiguous and the conclusions drawn from them are unambiguous. However, the conclusions are drawn on the basis of non-existent spots (lack of activities). The figure therefore appears to be presented somewhat unhappily, as it only shows blue areas. The authors are encouraged to consider a different way of presenting these results.

Author Response

Our reply to the specific comments is as follows:

As pointed it out by the reviewer, the original Fig. 8 was not suitable to sufficiently explain our conclusion. Therefore, we prepared a new version of Fig. 8 and added the control experiments using the active enzymes, the CCN’ proteins that had been auto-processed from proCCN’ and [K8D]-proCCN’. In addition, an enzyme assay using Bz-Arg-OEt as a substrate was also added to make our results clear. In the casein zymography assay, the active enzyme is observed as a white band while the inactive enzyme is not observed on the blue gel.

We also added several sentences to explain the results of the experiments in Fig. 8 in the revised manuscript.